# An Ultra-Sensitive Multi-Functional Optical Micro/Nanofiber Based on Stretchable Encapsulation

**DOI:** 10.3390/s21227437

**Published:** 2021-11-09

**Authors:** Siheng Xiang, Hui You, Xinxiang Miao, Longfei Niu, Caizhen Yao, Yilan Jiang, Guorui Zhou

**Affiliations:** Department of Engineering Optics, Research Center of Laser Fusion CAEP, Mianyang 621900, China; xiangsiheng1230@163.com (S.X.); yhworkyx@163.com (H.Y.); miaoxx@caep.cn (X.M.); niulf@caep.cn (L.N.); yaocaizhen2008@126.com (C.Y.); jiangyilan1023@163.com (Y.J.)

**Keywords:** optical micro/nanofiber, multi-functional, flexible encapsulation, wearable healthcare sensor

## Abstract

Stretchable optical fiber sensors (SOFSs), which are promising and ultra-sensitive next-generation sensors, have achieved prominent success in applications including health monitoring, robotics, and biological–electronic interfaces. Here, we report an ultra-sensitive multi-functional optical micro/nanofiber embedded with a flexible polydimethylsiloxane (PDMS) membrane, which is compatible with wearable optical sensors. Based on the effect of a strong evanescent field, the as-fabricated SOFS is highly sensitive to strain, achieving high sensitivity with a peak gauge factor of 450. In addition, considering the large negative thermo-optic coefficient of PDMS, temperature measurements in the range of 30 to 60 °C were realized, resulting in a 0.02 dBm/°C response. In addition, wide-range detection of humidity was demonstrated by a peak sensitivity of 0.5 dB/% RH, with less than 10% variation at each humidity stage. The robust sensing performance, together with the flexibility, enables the real-time monitoring of pulse, body temperature, and respiration. This as-fabricated SOFS provides significant potential for the practical application of wearable healthcare sensors.

## 1. Introduction

At present, the research and application of flexible wearable electronics are reflected in various aspects of human life [1,2,3,4], such as electronic skin, wearable physiological monitoring and treatment devices, flexible conductive fabrics, thin-film transistors, and transparent thin-film flexible gate circuits [5,6,7,8,9,10,11]. The term ‘flexible wearable electronics’ generally refers to electronic devices or devices that have mechanical flexibility and can directly or indirectly adhere closely to the skin, including a group of functional devices that can respond to external stimuli via capacitive [12,13], resistive [14,15], piezoelectric [16,17], and triboelectric effects [18,19]. However, the need for higher sensitivity, faster response, and better anti-interference properties may push the limit of wearable electronic sensors due to the nature of low-frequency electromagnetic circuits. For example, drawbacks such as parasitic effects, and the risk of short circuit and electromagnetic disturbances, might restrict the practical applications of wearable electronics [20,21].

In addition to traditional electrical sensing, combining optical sensing with soft and stretchable packaging may provide an alternative means of developing cheap but reliable wearable sensors [22,23]. With respect to optical sensors, there are abundant attractive merits compared with conventional sensors, including ultra-high sensitivity, anti-electromagnetic interference, and excellent insulation, resulting in variation in the monitoring of environmental parameters [24]. For example, Nag et al. presented a new approach to the development of flexible transparent strain sensors which combined the use of a transparent fabric with a polydimethylsiloxane (PDMS) polymer using a layer-by-layer assembly process [25]. Yang et al. demonstrated, for the first time, a flexible optical fiber capable of sensing a wide range of body movements using the liquid PDMS-based tubular mold method [26]. Despite the success of these enlightening optical sensors, the intricate fabrication procedure of flexible and stretchable sensors makes commercial and practical usage difficult. As a combination of fiber optics and nanotechnology, optical micro/nanofibers (MNFs) have attracted increasing research interest due to their excellent properties, such as strong light field constraints, small bending radius, and high mechanical strength [27,28,29]. In particular, their strong evanescent fields ensure that MNFs are highly sensitive to tiny variations in the surroundings. It is worth noting that the as-fabricated MNFs, with their high performance and competitive scale, are vulnerable to the severe disturbances that occur in the application of wearable sensors. In order to employ MNFs in the development of reliable and robust stretchable optical fiber sensors (SOFSs), the packaging scheme must be well managed. For instance, Daniel et al. employed the change in the elastomer refractive index and modes confined within the fiber core when PDMS was pressed, fabricating a locally pressed etched optical fiber with a PDMS coating for sensing applications [30]. Additionally, Jesus et al. reported a compact and highly sensitive optical fiber temperature sensor based on the surface plasmon resonance effect, showing a linear response and a sensitivity of 2.6 nm/°C [31]. Despite the high performance of the abovementioned sensors, a simple-to-fabricate, compact, hypersensitive, and multi-functional flexible optical fiber sensor remains a challenge for MNFs to some extent.

To address this issue, we propose a method to fabricate multi-functional stretchable fiber sensors by using a PDMS membrane embedded with MNFs. Based on the transition from guided modes to radiation modes of the waveguiding MNFs upon external stimuli, the as-fabricated MNF sensors are demonstrated to have outstanding sensitivity and repeatability when used in strain and temperature sensing. Moreover, by taking advantage of the water-absorbing ability of tin oxide, a wide detection range and highly sensitive humidity sensing are realized.

## 2. Materials and Methods

### 2.1. Fabrication and Manipulation of MNFs

In order to fabricate the MNFs, we used an electricity-heated mechanical stretching technique to draw a standard silica optical fiber into a biconical tapered fiber. In this method, a 3 cm-long section of the standard fiber was stripped of its cladding layer and fixed onto the MNF translation platform. When the fiber was heated to an optimal temperature, it was drawn into the horizontal plane until the diameter was reduced to the desired value. The entire MNF fabrication process was controlled by a computer program. To obtain uniformity among MNFs with different diameters, the temperature distribution in the drawing region and the speed of the stepper motors could be adjusted.

### 2.2. Fabrication of an SOFS

To fabricate an SOFS, we developed a sandwich-like structure to encapsulate the as-fabricated MNFs, which involved a three-step process: (1) 25 um of PDMS was prepared on a thin polyethylene (PET) flake to support the MNFs, which was firmly held in position using van der Waals forces and electrostatic interactions between the MNFs and the PDMS membrane; (2) 1 mL of degassed PDMS (mixing ratio of PDMS resin and curing agent was 10:1, Dow Corning Sylgard184) was slowly poured onto the PDMS/PET substrate, waiting for several seconds to ensure the substrate was covered with the fluid PDMS and that the MNFs were enclosed in the liquid; and (3) another thin PDMS/PET flake was placed on the substrate, followed by curing at 80 °C for 30 min, forming a 500 um-thick PDMS thin film. As for humidity sensing, a humidity-sensitive layer (tin oxide) was deposited using the magnetron sputtering method on the as-fabricated MNFs; the thickness of the tin oxide layer was about 20 nm for the sake of optimal sensing performance. According to the requirement of various sensing applications, the as-fabricated SOFS was peeled from the PET sheet.

### 2.3. Sensing Process of SOFS

Computer-controlled stepper motor translation platforms, which precisely control different step movements (including single step movement and periodic step movement), were prepared to measure the strain sensing performance of the SOFS. Before strain sensing, the as-fabricated SOFS was fastened onto the translation platforms with standard clamps and coupled with a 1550 nm wavelength fiber laser at one end. The optical output was detected by a spectrometer at the other end. The sensing process can be divided into two main parts: the first is a micro-strain measurement which contains several micron dimension step movements, and the other is an endurance test with hundreds of cycles of movements. Finally, in order to demonstrate the capability for health monitoring, the detection of cardiopalmus was conducted.

A constant temperature heating facility was used to test the temperature sensing performance of the SOFS, which could provide a stable temperature environment (ranging from 30 to 60 °C). During the temperature sensing process, the as-fabricated SOFS was positioned within the facility at all times and was always in a closed environment. Moreover, in order to demonstrate the capability of the SOFS for health monitoring, the detection of body temperature was conducted. A humidity generator and a detector were prepared to support the measurement of the SOFS’s humidity sensing capability, which provided a continuous standard humidity atmosphere (ranging from 10 to 90% RH). Furthermore, for the duration of the humidity sensing process, the as-fabricated SOFS was placed in an airtight chamber. In order to demonstrate the capability for health monitoring, the detection of respiration was conducted.

### 2.4. Characterization Studies

An optical fiber laser (YMPSS-980-750-M-FBG, YM, Suzhou, China) and an optical fiber spectrometer (AQ6370D, YOKOGAWA, Japan) were used as a light source and a detector, respectively. A humidity generator (Su Zhou Hua Xiang Star Environmental Technology Co., Ltd., Suzhou, China) and a humidity detector (HC2A-DP, Rotronic) were used to generate standard humidity and measure the unknown environmental humidity, respectively. When a single-mode optical fiber (9/125, Corning, New York, NY, USA) was coupled with the fiber laser, the input optical power of the MNF was about 16 mW, with a stability of less than 0.05%. The stepper motors of the optical fiber drawing platform (V-508.932020, PI, Germany) and the translation stage (Zolix, China) were used to investigate the strain response of the sensor. A home-made system consisting of multiple channels was used for monitoring the response of the optical sensor to humidity, maintaining a relatively long recording period. A constant temperature heating facility (W-KW8-6-180, China) with a temperature resolution of 0.1 °C was provided to maintain a stable temperature in the range of 30 to 60 °C.

## 3. Results and Discussion

### 3.1. Concept and Principle of the SOFS

Although the conventional PDMS embedding method has been widely employed [32,33], demonstrating outstanding quality in the field of wearable sensors, there are still certain aspects of durability that should be considered in its application. Thus, in order to obtain a sufficiently robust SOFS, we added a pouring process to ensure that the SOFS was fastened firmly between the PDMS membranes. In this work, we employed a simple but dependable procedure to fabricate a sandwich-like SOFS—a schematic illustration is shown in Figure 1a, and detailed descriptions are displayed in the Materials and Methods section. Taking practical sensing objects into account, there may be little difference in the SOFS in terms of structure. Moreover, we chose PDMS to embed the MNFs because of its low refractive index (RI) compared with silica, flexibility, and biocompatibility. The stretchable PDMS film not only effectively fixes the MNFs and isolates the evanescent field but also maintains environmental stimuli transduced on MNFs with high sensitivity.

Experimentally, highly uniform MNFs are fabricated through the tapered drawing of silica glass optical fibers, which allows for a guiding light at the micrometer scale. However, the as-fabricated MNFs, with a large proportion of evanescent fields exposed to the surroundings [34,35], are highly susceptible to environmental disturbance (e.g., pressure, vibration, and RI change) or contamination (e.g., dust and organic conglutination), which may lead to unpredictable perturbation of the guided signals. With the protection of the stretchable PDMS film, the above problems are able to be effectively suppressed to some extent, and an SOFS can be attached to the skin (Figure 1b) or pasted onto a glass cylinder with quite a large angle of bending (Figure 1c), exhibiting flexible and reliable characteristics in terms of wearable optical sensors.

Bending a flexible SOFS embedded with PDMS can lead to a bending-dependent transmission, which includes large-scale winding and tiny vibrations, enabling strain sensing. The RI of PDMS is a function of the temperature due to its large negative thermo-optic coefficient (about −1.0 × 10^−4^ per °C), and thus the guided modes of the MNFs are sensitive to the surrounding variation of RI owing to the large fractional evanescent fields outside the MNF, ensuring the SOFS’s responsiveness to the change in temperature. Moreover, a wider expansion of aspects of sensing media could be compatible with the as-fabricated SOFS. Tin oxide, as a humidity-sensitive material, has already been used in humidity sensing [36,37,38,39,40,41]. Combining sensitive materials and MNFs embedded with PDMS could be a potential application in wearable sensors, such as in breathing masks and pulse monitors. Thus, the stretchable and flexible as-fabricated optical sensor offers promising capabilities for strain, temperature, and humidity sensing.

### 3.2. Strain Sensing

To investigate strain sensing, firstly, the SOFS embedded with a 1 um-diameter MNF was fastened to a computer-controlled stepper motor translation platform for bending sensing. By taking advantage of the high-precision moving stage, we designed each step to correspond to a replacement of 1 um, 2 um, and 5 um with 10 times bend and 10 times recover. As shown in Figure 2a–c, the MNF embedded with PDMS was manipulated using a micro-shift for strain response, which showed an approximately linear relationship between the output intensity and the applied strain (various degrees of bending). In order to exclude the disturbance of the light source and other factors, the displayed curves were detected using an optical fiber spectrometer at 1550 nm, which is the center of the wavelength of the light source, and where the lowest optical loss of such a single-mode optical fiber occurs. As for the 1 um per step test, with 10 times bend and recover movement, the response curve indicated a good reversibility of the sensor. Moreover, by connecting the points of the average data from every flat stage, the results present a highly linear relationship between the output intensity and the applied bending range. The relative intensity change ∆*I*/*I*_0_ (*I*_0_ defined as the original output intensity of the SOFS) as a function of strain leads to a gauge factor (GF), where the gauge factor is defined as GF = (∆*I*/*I*_0_)/strain. Meanwhile, the detectivity of the SOFS was shown to be highly sensitive, with a gauge factor of 118 and higher compared to conductive core–shell aramid nanofibrils (GF:18.8) [42], and silver nanowire elastomer nanocomposites (GF:14) [43]. Considering the fact that the limit of the spectrometer is 0.01 dBm, it is worth noting that nanometer-scale bend or micro-strain detection could be achieved. Compared with the above test, the 2 um and 5 um per step tests also manifested similar variation trends with a better response (GF:205 and GF:450, respectively), though with slightly weaker linearity, which is close to the recorded value (GF:1000) in cracking-assisted strain sensors [44]. To demonstrate the diameter optimization of the as-fabricated strain sensor, 3 um and 5 um-diameter SOFSs were used for the strain measurement, as shown in Appendix A. With regard to the 3 um-diameter SOFS, the response to strain (GF:64) was weaker than that of its 1 um counterpart, while the the 5 um-diameter SOFS was even more insensitive (GF:44), exhibiting no obvious response below a 20 um displacement. Thus, considering the practicability and sensitivity, the 1 um-diameter SOFS exhibited the best performance, showing potential for further research.

The long-term reversibility and stability of the SOFS were further investigated by automatically bending and recovering the sensor over 250 cycles using a computer-controlled moving platform. After the cycles, the sensor maintained a steady baseline intensity with a slight decrease of 5% (Figure 2d), which might be attributed to the hysteresis effect of PDMS or an error of the moving platform. In order to carry out a thorough inquiry of durability, more bending and recovering cycles were performed. The responses from approximately 1000 cycles are shown in Appendix A: with an increase in repeats, the response intensity of the SOFS decreases gradually. This attenuation of the intensity might be caused by the variation in the elastic property and/or a change in the refractive index of the PDMS membrane. To demonstrate the capability for health monitoring, the SOFS was attached to the wrist of a volunteer for the detection of cardiopalmus, which is a crucial aspect of physiological sensing. As shown in Figure 2e, compared with the contrast curve (maintaining the same bending level as the wrist), the response curve of the heartbeat exhibits a rapid fluctuation with a calculated 68 beats per minute.

### 3.3. Temperature Sensing

In addition to strain sensing, the refractive index characteristics of PDMS are promising for temperature sensing; it has a large negative thermo-optic coefficient, while the silica MNF shows a negligible change compared with that of PDMS. The SOFS with a 1 um-diameter MNF embedded in a 500 um-thick PDMS film was placed on a computer-controlled constant temperature heating facility, which provided a stable temperature in the range of 30 to 60 °C. The transmission spectra of the SOFS at different temperatures were recorded using a spectrometer (Figure 3a). The process of increasing the temperature was manipulated by a computer, which included 12 parts in all (part 1 increased the temperature from 30 to 35 °C, part 2 held the temperature at 35 °C for 90 s, part 3 increased the temperature from 35 to 40 °C, part 4 held the temperature at 40 °C for 90 s, and the rest were carried out in the same manner). With an increase in the temperature, the response curve indicated an approximately linear relationship between the output intensity and the applied temperature range. By defining the temperature sensitivity as S = ∆*I*/∆*T*, where ∆*I* is the increase in the output intensity, and ∆*T* is the change in the temperature, the sensor achieved a sensitivity as high as 0.02 dBm per °C, which is similar to that of wearable temperature sensors based on metal oxides [45]. The decrease in transmission loss showed a tighter confinement of the guided light caused by the increased refractive index contrast between the PDMS and the MNF. In order to demonstrate its use as a wearable temperature sensor, the SOFS was directly pasted onto the back of the hand for temperature monitoring (the inset of Figure 3b). In order to avoid any disturbance caused by body motion, the subject tried to keep his hand as still as possible. To rule out changes in the ambient temperature, the whole process of measurement was conducted under a standard hundred-grade clean laboratory, maintaining a temperature of 25 °C and 50% RH all year. Figure 3b presents the typical response of the SOFS to a constant body temperature with remarkable stability, and no obvious change was observed during the period of detection.

### 3.4. Humidity Sensing

Besides the excellent stretchability and temperature response, an ultra-sensitive optical sensor with a wide range of humidity detection was obtained according to the Materials and Methods section. Interestingly, when we started to study the effect of different diameters on the humidity response, SOFSs with diameters that were too small or too large under the same thickness of the sensitive layer were demonstrated to be unresponsive. The phenomena of guided light scattering and optical absorption (conversion from light to heat, as shown in Appendix A) caused by the tin oxide layer are the main possible reasons for this; thus, balancing the MNF’s parameters and the thickness of the sensitive layer appears to be of great importance to the humidity response. To investigate the humidity response, the SOFS embedded with an optimal 2 um-diameter MNF was placed in different humidity atmospheres, which was tested in an airtight cavity with a humidity generator and an electronic humidity detector.

Compared to the above sensors, the pattern of encapsulation was discriminating to some extent, which included a spindly silt (less than 0.5 cm^2^) close to the most sensitive area in order to ensure wet air penetrated into the humidity-sensitive material. Tin oxide, a promising humidity-sensitive material, has not only been widely used in the field of humidity sensing but also popularized in the field of opto-electronics for decades [46,47,48]. It is worth noting that the morphology of tin oxide is characterized by its porous structure, which is beneficial for interacting with vapor and allowing vapor to transfer more smoothly. After combining with water vapor, the refractive index of tin oxide will change, which will affect the evanescent wave transmitted outside of the fiber as well. Figure 4a presents a continuous humidity measurement ranging from 10% to 90% RH using a home-made monitoring facility. At 10% to 30% RH, the as-fabricated sensor shows an inconspicuous response, probably due to the lack of enough water reacting with the tin oxide. Moreover, with the increase in humidity, the SOFS exhibits an excellent humidity sensing capability, especially at high concentrations of wet air (above 60% RH). Note that the robust sensitivity of the humidity sensor offers an opportunity for practical applications in daily life. By defining the humidity sensitivity as S = ∆*I*/∆*H*, where ∆*I* is the increase in the output intensity, and ∆*H* is the change in the relative humidity, the as-fabricated sensor exhibits a peak sensitivity of 0.5 dB/% RH, which is far better than a humidity sensor based on a hetero-core optical fiber [49] and a no-core optical fiber [50], and an approach to humidity sensing based on a core-offset fiber Mach–Zehnder interferometer (S = 0.104 dB/% RH) [51]. Considering the durability of such a sensor, we also proceeded with more tests to demonstrate its repeatability, as shown in Figure 4b. A moderate fluctuation in the intensity response at every humidity stage was gauged with less than 10% variation. In particular, there is a better performance of the response within three times repeated measurements, which exhibits a flat curve (with less than 3% variation) of the output intensity except for the high-humidity conditions (90% RH). With the increase in repeated measurements, because continuous humidity testing from dry to wet might make the sensitive material saturated to a certain degree, the fluctuation in the intensity response becomes more or less larger under each of the humidity conditions.

To exclude the disturbance of continuous humidity measurement and demonstrate the effect of the SOFS under all humidity conditions, the humidity stages from 30% to 90% RH were conducted individually. As shown in Appendix A, in this case, the holistic response tendency maintains that of its continuous counterpart. Moreover, under high-humidity conditions (greater than 60% RH), the output response displays an approximately symmetrical humidity and recovery cycle, while the recovery response at less than 60% RH shows a slightly weaker cycle, which might be attributed to the residual water penetrating into the porous tin oxide. To function as a wearable humidity sensor, the SOFS was directly integrated with a mask for the detection of respiration (Figure 4c), which is also a crucial indicator in the field of health monitoring. As shown in Figure 4d, under deep breath conditions within the breath-monitoring mask (12 min^−1^), the humidity response curve presents quite a sharp peak and an excellent reverting capacity. In addition, the as-fabricated SOFS possesses a better response speed (of a few seconds) compared with its electronic counterpart (of minutes, especially for the recovery response) under a wide humidity range.

## 4. Conclusions

In summary, we have demonstrated an ultra-sensitive multi-functional optical micro/nanofiber embedded with a flexible encapsulation of PDMS, which possesses great potential in the field of wearable optical sensors. By taking advantage of the character of its strong evanescent field, the as-fabricated MNF embedded with a PDMS membrane is highly sensitive to weak bends, enabling a robust durability within hundreds of measurements and high sensitivity with a peak gauge factor of 450. At the same time, taking the large negative thermo-optic coefficient of PDMS into account, a temperature response in the range of 30 to 60 °C was realized, resulting in a resolution of 0.02 dBm/°C. In addition, a humidity measurement within a broad detection range from 30% to 90% RH was demonstrated, with a sensitivity of 0.5 dB/% RH, and less than 10% variation at each humidity stage. Moreover, the as-fabricated highly sensitive sensing performance with flexibility enables the real-time monitoring of pulse, body temperature, and respiration. The above results may pave the way towards future wearable optical sensors for health monitoring including real-time monitoring of pulse, respiration, and other aspects of life signs.

## Figures and Tables

**Figure 1 sensors-21-07437-f001:**
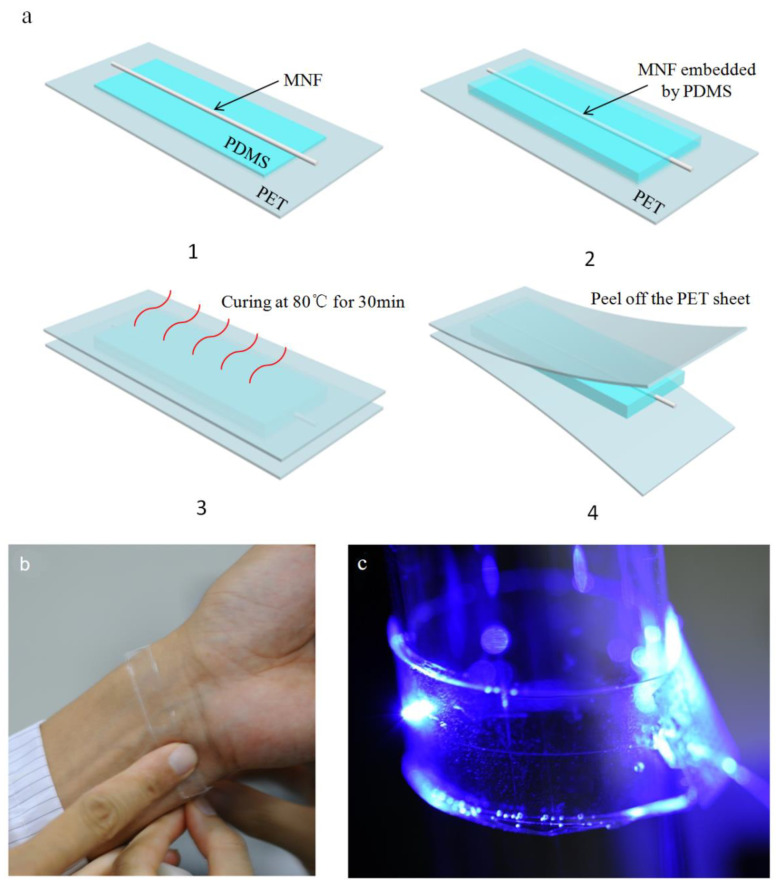
(**a**) Schematic illustration of the fabrication process of the sandwich-like SOFS. (**b**) Photograph of an SOFS attached to the wrist. (**c**) Photograph of an MNF-embedded PDMS patch pasted onto a glass tube.

**Figure 2 sensors-21-07437-f002:**
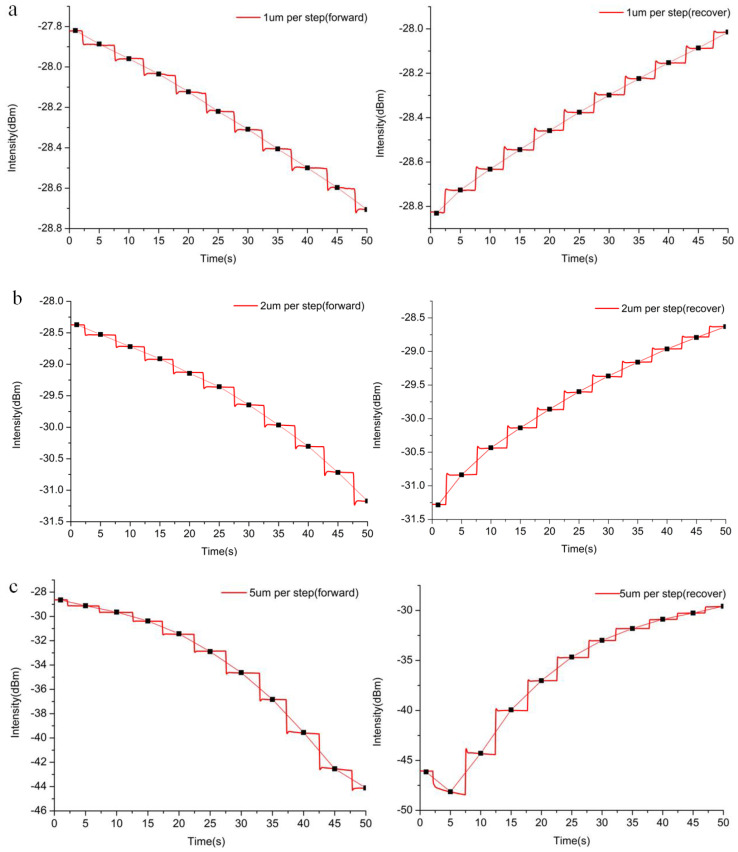
Characterization of the SOFS for strain sensing. Changes in optical intensity output derived from bend and recover tests. Each step corresponds to a displacement of (**a**) 1 µm, (**b**) 2 µm, and (**c**) 5 µm. (**d**) The durability test under a bend of 100 µm at a frequency of ~0.5 Hz. Inset: a magnifying view of the part of the response curve after 75 bend–recover cycles. (**e**) Measurement of the wrist pulse under normal conditions (about 68 beats per minute) and contrast conditions (maintaining the same bending extent without pulse).

**Figure 3 sensors-21-07437-f003:**
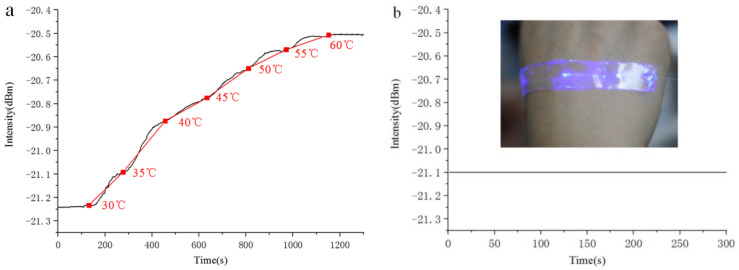
(**a**) Continuous temperature response of an SOFS measured in the range of 30–60 °C. (**b**) Real-time measurement of body temperature by attaching an SOFS to the back of the hand. Inset: a photograph of an SOFS attached to the back of the hand, which guided the 405 nm-wavelength laser for visualization.

**Figure 4 sensors-21-07437-f004:**
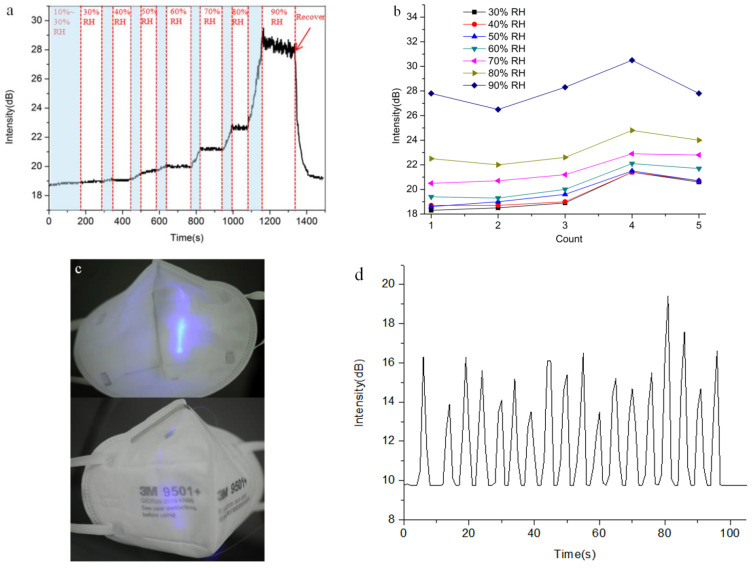
Characterization of the SOFS for humidity sensing. (**a**) Continuous humidity test ranging from 10% to 90% RH; the light blue area is the process of increasing the humidity. (**b**) Fluctuation in the intensity response at every humidity stage corresponding to five repeated measurements. (**c**) Photograph of a home-made breath-monitoring mask. (**d**) Real-time measurement of the respiration of a volunteer under deep breath (12 min^−1^) conditions.

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
