# Peer review of "An Ultra-Sensitive Multi-Functional Optical Micro/Nanofiber Based on Stretchable Encapsulation"

_sensors, 2021, doi:10.3390/s21227437_

Round 1

Reviewer 1 Report

This work reported a micro/nanofiber embedded with PDMS flexible membrane, which is well compatible with wearable optical sensing. The work is interesting and effective. However, there are still some shortcomings that need improvement. I suggest publishing after major revision.

Comments:

  1. The detailed experimental process of the strain/temperature/humidity sensing of SOFS needs to be explained in the experimental part of the manuscript. Please add it.
  1. I suggest establishing a corresponding standard curve for strain, temperature and humidity sensing to prove sensitivity and quantification of SOFS.
  1. When SOFS works as a temperature sensor, is the temperature of the human body or the temperature of the environment detected? How to make sure that the sensor is not affected by the environment?
  1. When performing temperature sensing tests, why disturbance caused by body motion can be neglected? Can the interference caused by the moved hand skin be eliminated?
  1. The work lacks optimization of some conditions, such as the width of the optical fiber, the thickness of the PDMS, etc. Please add it.

Reviewer 2 Report

In this manuscript, the author has designed a simple MNFs based SOFS sensor for humidity sensing, temperature sensing, and strain sensing. Some issues need to be addressed before publication.

  1. Please explain the humidity sensor mechanism. How does tin oxide work in the humidity sensor?
  2. Is there any diameter optimization experiment for the strain sensor? Why 1 micrometer?
  3. Page 7, line 235, "2" should be superscript.
  4. The response time of the humidity sensor is almost 60 seconds to reach the equilibrium. Has the author tried any method to decrease the response time?
  5. Figure 4b, are these seven curves obtained from the same device? Why is there a peak at the 4 counts?
  6. Figure 4a, from 10% to 80%, the intensity reading increases from ~19 to ~23. It looks like the sensitivity is not relatively high. This low sensitivity may affect the application of this sensor. Any explanation from the author?

Reviewer 3 Report

The reviewed paper is about stretchable fibre optic sensors used for health monitoring. The paper is very interesting and fully meets the requirements for the journal SENSORS. However, it contains several shortcomings among which are: 
1. abbreviations should not be used in the abstract, if they must be used the full name should also be given
2. the theoretical introduction should be supplemented with other classes of sensors and biosensors used in medicine. In this case the following papers should be considered: 10.1109/JSEN.2020.2982742
3 The authors claim to have obtained an ultra-sensitive sensor, but do not compare with literature data. This needs to be supplemented
4. please very much consider the chemical composition of the obtained sensors, and what chemical level interactions are responsible for the analytical parameters obtained. 

Round 2

Reviewer 1 Report

The reviewer appreciates the author's careful and sincere revision of the manuscript. All concerns raised in the first review have been adequately addressed.

Reviewer 2 Report

The revised version has addressed my questions. Thanks.

Reviewer 3 Report

The authors made corrections according to my suggestions, so I recommend publication of the reviewed paper.